



# Oceanographic preconditions for planning seawater heat pumps in the Baltic Sea – an example from the Tallinn Bay, Gulf of Finland

Jüri Elken[1], Ilja Maljutenko[1], Priidik Lagemaa[1], Rivo Uiboupin[1], Urmas Raudsepp[1]

[1]Department of Marine Systems, Tallinn University of Technology, Tallinn, 12618, Estonia

*Correspondence to*: Jüri Elken (juri.elken@taltech.ee)

**Abstract.** The use of low temperature seawater heat for renewable energy installations is demonstrated with an example from the Tallinn Bay, Baltic Sea based on Copernicus Marine Service reanalysis data. Tallinn and its surrounding seaside counties inhabit about half a million people and produce about half of Estonia's Gross Domestic Product (GDP). The Tallinn Bay with an area of 223 km$^2$ extends to the north and has an open connection to the Gulf of Finland. Depths more than 50 m, that cover

the halocline, appear already at a distance of 3–4 km from the coast. Surface layers get too cold during winter to be used in heat pumps for district heating; therefore, a feasible option is to pump slightly warmer seawater from the deeper halocline layers. The lowest monthly mean halocline temperature – down to 2.6 °C at 50-m depth and 3.3 °C at 70 m – is found in March and April, based on Copernicus Marine Service reanalysis data from 1993-2019. Seawater becomes less than 3 °C on the average on 1 January at 20 m depth and on 12 February at 50 m depth. At the 70 m depth, the average start of T < 3 °C was

calculated 28 February, although only 14 winters out of 26 had such water present; in 12 winters there was always the condition T > 3 °C fulfilled. Median number of cold days is 11, with a maximum of 128 days in a winter 1993/1994 when stratification became rather weak due to the prolonged absence of Major Baltic Inflows of saltier and warmer North Sea waters. During the recent warmer period 2009–2019, the start of cold seawater period has been delayed on the average by 5–10 days. Tallinn has among other Baltic Sea cities and industrial sites a favorable location for seawater heat extraction, because of the short distance

to the unfreezing sub-halocline layers. Still, episodically there are colder water events T < 3 °C, when seawater heat extraction has to be complemented with other sources of heating energy.

## 1 Introduction

New developments of district heating (e.g. Lund et al., 2018) focus on the gradual exclusion of fossil fuels and transfer to the

100% renewable energy sources. Among the latter, large-scale heat pumps are considered as an important component of the new energy systems. One energy source for the heat pumps is seawater (Bach et al., 2016). Regarding other studies, Su et al (2020) performed a spatial evaluation of seawater-source heat pump performance along the coast of China depending on their technical options and the thermal regime of seawater or other heat source (wastewater, groundwater etc.). Pieper et al. (2019) evaluated combinations of heat pumps, based on the data from Copenhagen and noted that in shallow water conditions seawater



heat may contribute 14% of the total heating energy. An important issue in coastal waters of ice-prone seas is to avoid lowering
of temperature in the heat pump system below the freezing temperature that depends on the salinity. Water temperature during
winter usually increases at depth, and hence, even depths of a few tens of meters may be sufficient to avoid freezing
temperature. One of the world's largest seawater heat pumps, built in Ropsten of Stockholm, is designed to work with the
lower limit of input water temperature +2.5 °C (Friotherm, 2017). Because of the large volume flows of water pumped from

the 15-m depth, cooling the water down to +0.5 °C yields in total 180 MW from 6 units. The study by Volkova et al. (2022b)
summarizing geographical and economic factors concluded that among the Baltic countries, seawater is the best natural heat
source in Estonia, but river water has a higher potential for the other two countries. Contemporary housing also needs district
cooling during the summer that can be performed based on the seawater (Volkova et al., 2022a).

Tallinn, about half-a-million inhabitants seaside city on the coast of the Gulf of Finland, Baltic Sea, is looking forward to using
seawater heat for district heating. Unfortunately, during the winter period, i.e. when the highest demand for heating the housing
exists, coastal seawater at shallow depths may cool down close to the freezing temperature, which may hamper the efficiency
of using the heat pumps. It is known from the larger Gulf of Finland area that during summer the mean halocline depth is
located at about 67 m, keeping the temperatures in the range of 2.2–5.0 °C depending on the transport of deep, more saline

waters from the open Baltic Sea (Liblik and Lips, 2011). The waters in the halocline and below are rather isolated from direct
vertical heating and cooling, therefore their temperature is more stable than in the surface layers. As systematic analysis of the
near-bottom water temperature in the Tallinn Bay is missing, the present study focused on the exploration of the new public
data sets made available by the EU Copernicus program. Namely, the Copernicus Marine Service
(https://marine.copernicus.eu/) has provided regular Baltic Sea reanalysis data during 1993–2019 with about 3.7 km grid step,

combining both modeling and observations (Table 1, product ref. no. 1).

The aim of the present study is to determine the basic features of seawater temperature variability, necessary for the evaluation
of the feasibility and efficiency of using seawater thermal energy for heating and/or cooling of the city built environment.
Presentation of data and methods is followed by providing results and discussions with focus on subsurface temperature

variations and state discussion with respect to the cold water periods. The paper ends with recommendations for further studies
and conclusions.

## 2 Data and methods

### 2.1 State of knowledge on oceanographic conditions in the Baltic Sea

The Baltic Sea is a multi-basin semi-enclosed sea (Fig. 1a) filled with brackish waters. Its temperature regime (Leppäranta and
Myrberg, 2009) is to a great extent forced by the seasonal cycle of atmospheric heat fluxes that extend down to the thermocline
during summer and to the halocline during winter (Fig. 1b). In halocline and deeper layers, temperature is determined mainly





by the deep-water transport of the waters originating from the North Sea (Leppäranta and Myrberg, 2009). However, these waters are significantly transformed on their pathway from the Danish Straits through the Baltic Proper to the Gulf of Finland.

(a)                                                                                       (b)

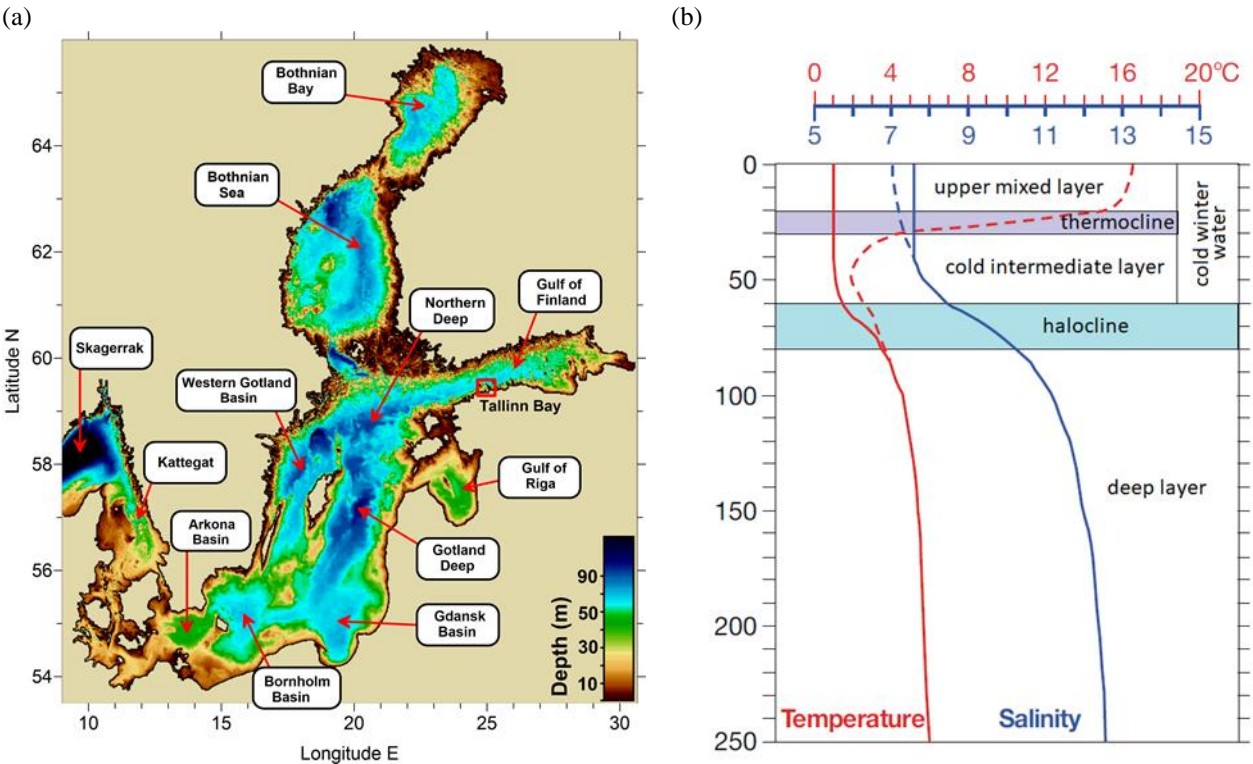

**Figure 1: Topography and basins of the Baltic Sea (a). Baltic Proper includes Arkona Basin, Bornholm Basin, Gdansk Basin, Gotland Deep and Western Gotland Basin. Typical vertical stratification in the Gotland Deep (b, adopted from Elken and Matthäus, 2008) in summer (dashed lines) and winter (solid). Location of Tallinn Bay is shown on (a) by a red box.**

While temperature and sea ice respond rapidly to changes in atmospheric heat fluxes, variations in salinity are governed mainly by lateral transport processes, resulting together with diapycnal mixing (i.e. mixing across the surfaces of constant water density) in response times of many decades (e.g. Omstedt and Hansson 2006; Elken et al., 2015). The Gulf of Finland is an elongated sub-basin of the Baltic Sea, which has many specific features (Alenius et al., 1998). Stratification is generally composed from the following layers:

(1) Upper mixed layer. Generally, the upper mixed layer is vertically rather uniform due to high mixing rates generated by winds, waves and convection during cooling (Leppäranta and Myrberg, 2009). During summer, it typically has a thickness of 20 m in the Gulf of Finland (Liblik and Lips, 2011). On calm and sunny days, a thin warm "skin" layer may occur just at the surface. However, with increasing wind speed the whole upper layer undergoes overall mixing again. The yearly temperature maximum occurs in July or August depending on the region and yearly atmospheric conditions (Haapala and Alenius, 1994).

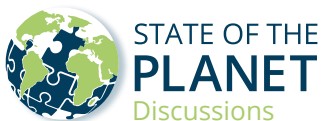

During the autumn cooling period, the upper layers become hydrostatically unstable. If cooled water with higher density
appears on top of warmer and less dense water, then instability generates convective mixing (Maljutenko and Raudsepp, 2018).
As a result, mixed layer temperature decreases but its thickness increases, until its lower boundary has eroded down to the
halocline or to the bottom, whichever is shallower. Shallow sea areas warm and cool faster during spring and autumn (also
called differential heating), respectively, because of the smaller water mass of the water column compared to deeper ocean
areas.


Cooling of the upper layer is limited during winter by the salinity-dependent freezing point of seawater. For salinity values of
5, 6 and 7 psu, the freezing point is -0.27, -0.33 and -0.38 °C, respectively. The salinity-dependent temperature of maximum
density $T_{max}$ influences upper layer dynamics during the cold period as well. For freshwater, $T_{max} = 4$ °C and for typical Baltic
surface salinity 7 psu $T_{max}$ is about 2.5 °C.


(2) Thermocline. During summer, a sharp temperature drop at depth develops below the upper mixed layer. Individual instant
profiles may exhibit sharpest thermoclines with temperature drops from 5 to 10 °C per 5 m (Fig. 1b), and the mean summer
thermocline thickness in the Gulf of Finland is 14 m (Liblik and Lips, 2011). With respect to the mean seasonal cycle, the
thermocline may perform up- and downward motions with periods from a couple of hours to several days. Vertical movement
of the thermocline of 10 m or even more creates significant near-bottom temperature variations. A special dynamical feature
of the thermocline is upwelling, when cold waters (sometimes below 5 °C) may be found at the surface of coastal areas during
summer (Aavaste et al., 2021). In the Gulf of Finland, upwelling occurs near the Estonian coast during easterly winds and near
the Finnish coast during westerly winds (Uiboupin and Laanemets, 2009).

(3) Cold intermediate layer. Cold waters from the last winter remain below the thermocline trapped through the whole summer,
until autumn cooling pushes the thermocline significantly downwards. In the Gulf of Finland, lowest summer temperature
(from 1.3 to 3.6 °C) has been found on average at 42 m depth (Liblik and Lips, 2011).

(4) Halocline and deep layers. This layer is formed by lateral (horizontal) transport of saline waters originating from the North
Sea. Regular saline water inflow is complemented by the events of Major Baltic Inflows (MBI) of large volume and high
salinity, they occur sporadically with interval from years to decades (Raudsepp et al., 2018). Depending on the seasonal timing
of the MBI, there may be pulses of warmer or colder water reaching the deep areas of the Baltic Proper (Elken et al., 2015).
For example, in the Gotland Deep at the 175 m depth, temperature varied during 1997–2013 between 4.5 and 7.0 °C depending
on the type of the MBI. When large volumes of new water arrive to the Gotland Deep, the old water is pushed to the deep
layers of the Gulf of Finland. After the 2014 December MBI, the first effects occurred in the Gulf of Finland in 9 months,
whereas the arrival of the former Northern Baltic Proper deep layer water was observed (Liblik et al., 2018). Deep layer



temperature dynamics in the Gulf of Finland is also affected by the wind-dependent estuarine circulation reversals that may drastically reduce the stratification for several weeks (Elken et al., 2003).

From May to July, the seasonal cycle of temperature in the central part of the Gulf of Finland shows delayed

temperature increase of the layers 20– 40 m during the warming season (Haapala and Alenius, 1994). The seasonal

upper layer temperature maximum develops usually in August. In the course of the cooling period, the thermocline deepens to

30 m (i.e., temperature of that level becomes equal to surface temperature) in September and 40 m in October.

The 50-m depth level lies in the cold intermediate layer at temperatures below 8 °C through the year. Average

temperature in the halocline (60– 90 m) is stable, ranging from about 3 °C in February and March to 5 °C in

November and December.

Sea surface temperature can be acquired in detail using regular satellite images, but subsurface data can be observed only using sparse in-situ techniques. Elken et al. (2015) summarized that during 1990–2008, sea surface temperature increased in the Gulf
of Finland at a mean rate 0.8 °C per decade. Liblik and Lips (2019) explored large amounts of observations conducted in the water column. They have shown that from 1982 to 2016 the temperature of the Gulf of Finland has been increasing on average at a rate about 0.5 °C per decade, whereas faster warming has been detected in the thermocline (20 m) and at deeper (70 and 80 m) depth levels. Re-inspection of temperature observations in the Gulf of Finland from the HELCOM/ICES database at 70 m depth level reveals that average deep-water temperature has increased from 3.5 °C in 1988 to 5.2 °C in 2020. Oceanographic
regional climate projections until 2100 have projected an annual increase of temperature in the Gulf of Finland 0.2–0.4 °C per decade (Meier et al., 2022).

**2.2 Data**

**2.2.1 Copernicus Marine Service reanalysis data for the Baltic Sea**

We use the Baltic Sea physical reanalysis product (Axell, 2021) prepared within the Copernicus Marine Service (product ref. no. 1). This reanalysis covers the whole Baltic Sea with adjacent North Sea areas. It has about 3.7 km grid step and it is the most advanced combination of numerical modeling made using the NEMO model, and the observations. The data set consists of daily mean reanalysis data from 01.01.1993 to 31.12.2019. The gridded data have maximum 56 vertical levels down to 361 m. Layer thickness is about 3 m in the upper levels, but increases to about 6.3 m in the deepest point of the Tallinn Bay area,
which has altogether 21 depth levels.



The quality of the reanalysis product was vigorously checked against observational data by Liu et al. (2019). The report concluded that the mean bias of water temperature was less than 0.1 °C and the remaining root-mean-square deviation (RMSD) was less than about 0.7 C. Comparing with individual independent observations that were not included into the reanalysis

procedure, the RMSD is between 0.4 and 1 °C. The largest RMSD values were found in the Kattegat and the Gulf of Finland. The grid cells of the NEMO model are shown in Fig. 2, overlaid on the coastline and topography map of the Tallinn Bay. Although depths 40–70 m are evident in the inner bay, the reanalysis grid cells of the Baltic-wide product are too coarse to resolve the details of coastline and topography. Setting up the more detailed topography for oceanographic reanalysis, dedicated to the Estonian marine areas, is in the implementation.


For the present analysis, data from 6 grid cells were selected. The data on the original grid were converted to 10-m depth intervals.

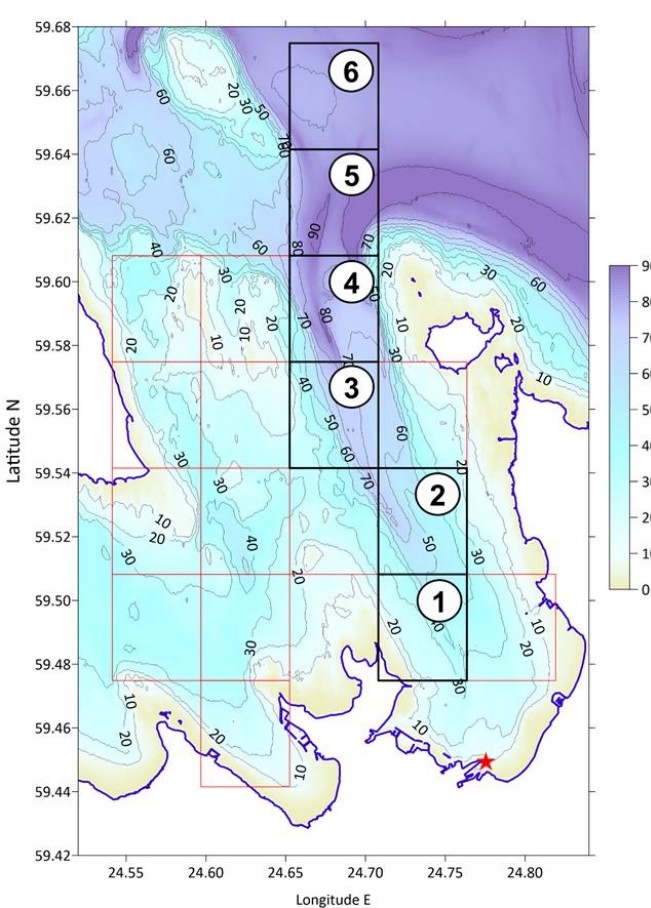

**Figure 2: Map of the Tallinn Bay area with grid cells of the Copernicus Marine reanalysis data (product ref. no. 1) shown as boxes. The analysis uses data from bold cells noted by numbers. Red star denotes location of the coastal automatic observing station. Depth**



**data on the 1/16 arc minutes (ca 58 and 116 m along longitude and latitude) grid were adopted from EMODNET bathymetry (product ref. no. 3). The coastline has been adopted from the compilation by HELCOM (product ref. no. 4).**

## 2.2.2 Observations


Regular temperature observations are available from the automatic coastal station (59.449351 N, 24.775448 E) located in Tallinn Harbor (Fig. 2) from 01.01.2008 to 14.04.2018 (product ref. no. 2). Sea level part of the observing station has been described by Lagemaa et al. (2011). The raw data have been recorded mostly with a 5 min interval. After elimination of sensor and communication errors, a set of daily mean temperatures was generated in order to compare with reanalysis data. The same

data, processed to the 1-hour interval, are available in the Copernicus Marine Service.

## 2.3 Comparison of reanalysis data with observations

While observational data record the "point" values of sea surface temperature with a 5 min interval and is usually averaged to hourly values, the reanalysis data set presents the average temperature over ca 3.7-km size grid cells as daily mean values. It means that reanalysis data do not present small-scale and short-term local temperature variations near the observation point.

Still it is interesting to compare how well point observations match with averaged reanalysis data. Point observations reveal a number of short-term anomalies from the regular seasonal course (Fig. 3). Their origins are in the processes: 1) calm weather local anomalies, when warm skin layer develops, mostly during spring, 2) faster heating and cooling in shallow coastal sites, 3) cold waters due to wind-induced upwelling. In all the cases, the spatial and temporal scales could be too small to be captured by the coarser reanalysis data. This study did not focus on a detailed analysis of "negative temperature anomaly" events, but

we can conclude that in many cases the observed events are evident also in the reanalysis data. Since the upwelling areas cover up to tens of km and they may exist for several days or a week (Uiboupin and Laanemets, 2009) then these events are captured well by the reanalysis data.

Statistical comparison of surface reanalysis data with daily mean observations over existing data pairs reveals their good match.

Bias is -0.12 °C and RMSD (root-mean-square difference) is 1.3 °C. Both time series contain strong seasonal signal, with its variance exceeding 90% of the total variance. While the original time series are highly correlated with Pearson $R^2 = 0.96$, then removing the mean seasonal cycle from both of the time series reduces the correlation to $R^2 = 0.44$. A histogram of the difference between observations and reanalysis (not shown) reveals that on average observations expose slightly higher temperature (the maximum is shifted by 0.2 °C), but occasionally much smaller observed temperatures occur. Such outliers

can be explained by the local coastal effects in observations, nor resolved by reanalysis.



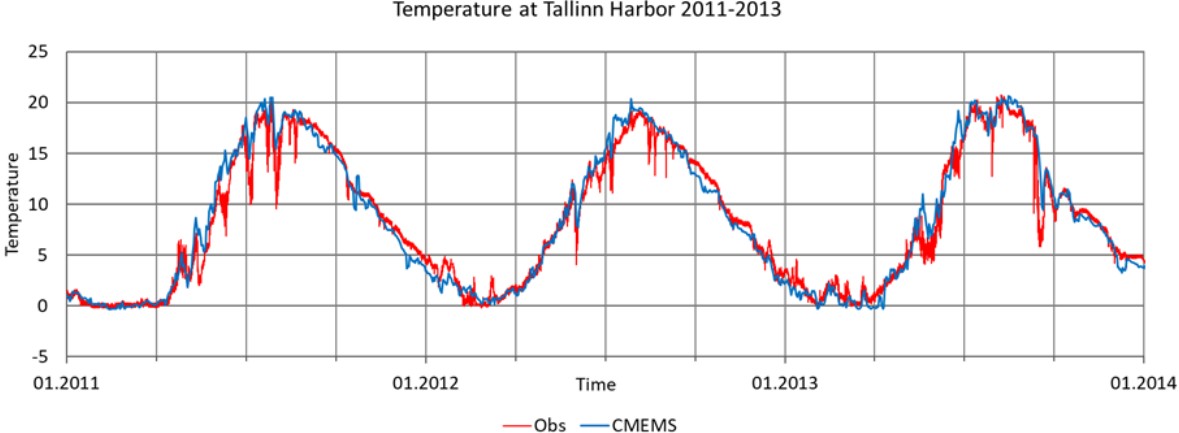

**Figure 3: Example of near-surface temperature time series in 2011–2013. Shown are the automatic observations at Tallinn Harbor with a 5 min interval averaged to hourly values (red) (product ref. no. 2) and daily Copernicus Marine reanalysis data in the point 1 (blue) (product ref. no. 1). For locations, see Fig. 2.**

## 3 Results

### 3.1 Seasonal temperature course and its variations

The temperature of the Tallinn Bay undergoes a similar seasonal cycle as described in Section 1. In the offshore area (point 6, Fig. 2) the upper layer is gradually warmed up from April to July (Fig 4a). During summer, the upper mixed layer has 10–20 m thickness, whereas the annual temperature maximum, determined from the original, non-averaged data, varies between 16 and 23 °C. In August, maximum vertical gradient in thermocline is found at about 25 m depth. During autumn, the upper layer and the thermocline are eroded down to 40 m where inherent salinity stratification (Fig. 4b) blocks further erosion. During winter, the upper layer may be cooled to the freezing point, which is from -0.27 to -0.33 °C for salinities from 5 to 6 psu. Below the thermocline, monthly mean temperature ranges from 2.4 to 7.4 °C at 40 m depth and from 3.4 to 5.2 °C at 70 m depth (Fig. 4a). Deeper temperature variations below 40 m are influenced by the seasonal dynamics of halocline: intensified transport of more saline (and warmer) waters from the open Baltic occurs below 50 m in late spring and summer (Fig. 4b). Highest temperature occurs in deeper layers in November and December as a delayed response to the summer heating at the surface; lowest temperature values are found on the average in March and April when there are the weakest vertical gradients in the halocline.





(a)

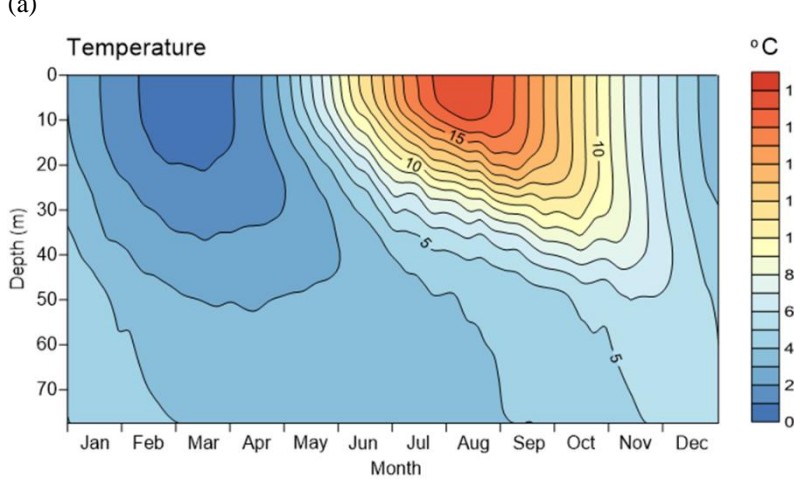

(b)

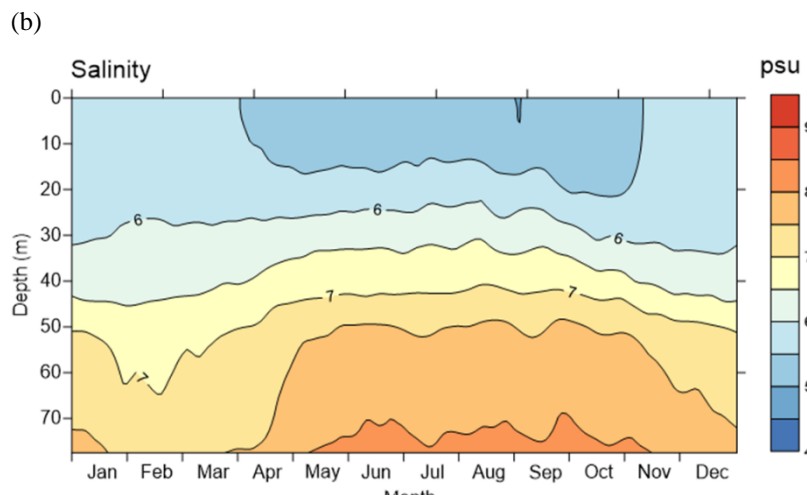

**Figure 4: Mean seasonal temperature (a) and salinity (b) in the Tallinn Bay reanalysis point 6 (Fig. 2) as a function of time of the year (month) and depth. Data from Copernicus Marine reanalysis 1993–2019 (product ref. no. 1).**

Although the mean seasonal temperature curves are rather similar in the selected six reanalysis points (locations in Fig. 2), the spread of the actual data is rather high (Fig. 5). Histograms of weekly mean temperature reveal during the summer months a range from 4.5 to 22.5 °C at the depth of 10 m, with a maximum in overall seasonal mean of 16 °C. The cold anomalies are rather rare – summer temperatures below 10 °C cover only 7% of occurrence frequency. During winter months, temperature variability is confined between -0.2 and 3.5 °C at 10-m depth, while at 50 m the limits are 0.5 and 5.4 °C. Mean seasonal data

from individual reanalysis points reveal minor horizontal variations between the points, compared to their temporal variations. For example, during the winter when deep seawater is needed for the heat pump, the pointwise monthly mean temperature varies at 50-m depth between 2.6 and 2.8 °C in March, but it has a mean seasonal maximum of about 5.9 °C in November and December. Detailed analysis reveals that temperature of deep layers below 40 m depth can be considered practically uniform





over the bay area on the scales of a few km and a few days, that is sufficient for the baseline evaluations of oceanographic

conditions for seawater heat pumps.

(a)

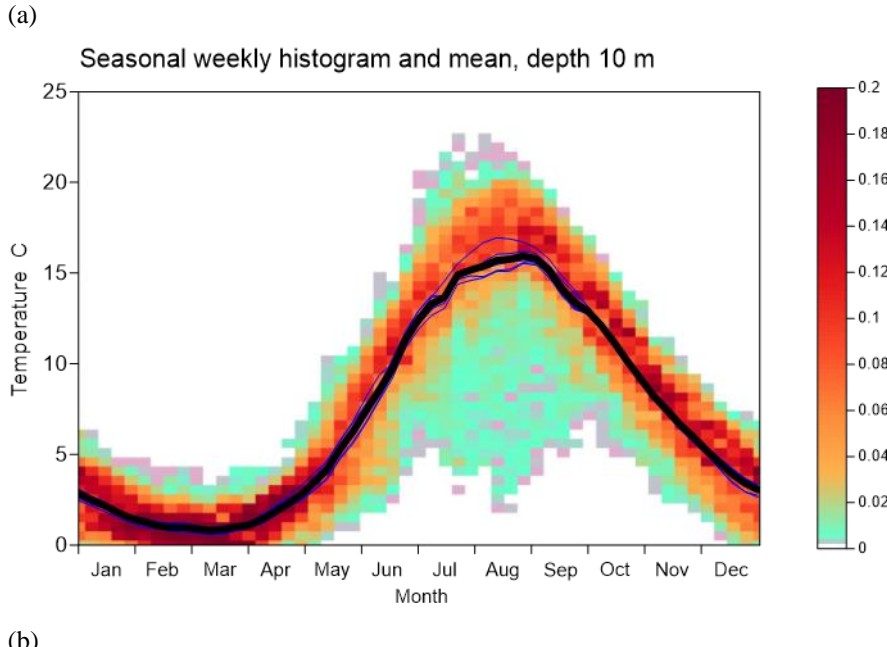

(b)

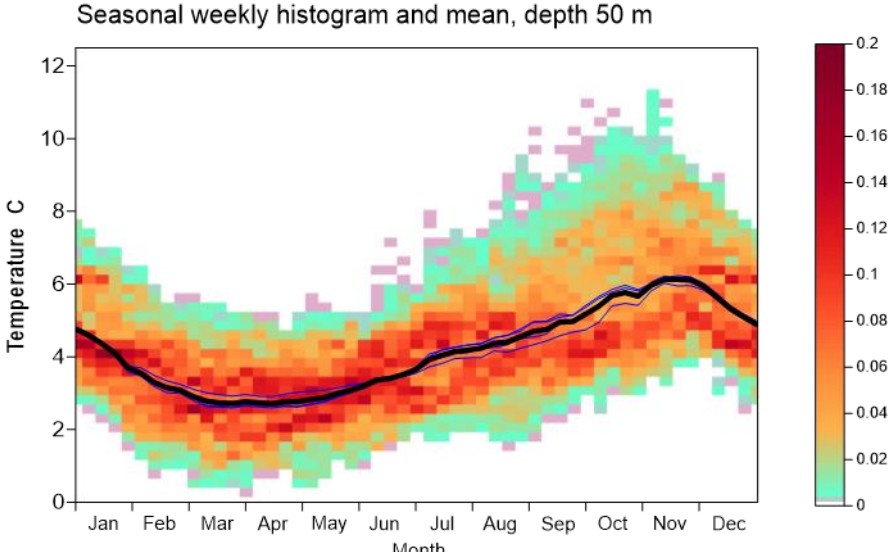


**Figure 5: Mean seasonal curve (black solid line) and seasonal temperature histogram (color image) in Tallinn Bay at depths 10 m (a) and 50 m (b). Shown are also mean temperature curves for individual horizontal points (thin lines). Copernicus Marine reanalysis data from 1993–2019 (product ref. no. 1) taken from six locations (Fig. 2) are processed with 7 day (weekly) intervals. The temperature histogram bins are 0.5 °C for 10 m depth (a) and 0.25 °C for 50 m depth (b). For each week, the sum of histogram**
**frequency shown by color scale is 1.**





## 3.2 Statistics of extractable seawater heat

The data were segmented into 26 yearly heating-centered periods, from 1 July to 30 June. For the data visualization of each year, a date-year column diagram was drawn, where the color stripes in the column correspond to the actual temperature in a
specific year, based on the defined color scale (Fig. 6). Results show that both on the 50 and 70 m level of the offshore point 6, the highest temperatures above 4.0 °C occurred every year, but the duration and timing is variable. During the winters 2000/2001, 2007/2008 and 2008/2009, deep layers were warmer than usual through the whole year. In several years, the warmer deep waters were dominant until February or March. There were also years, when colder waters were present in summer and extended to autumn, a time when deep-water temperature started to increase.


We define that seawater heat is extractable when water temperature exceeds some predefined value while cold water occurs when the temperature is below that limit. Cold period durations and start dates were determined by evaluating running 9-day slices of temperature criteria fulfilment in the time series. Shorter duration spikes with less than four fulfilments were considered as not fulfilled. For the heat pump problem, the time series were limited to periods from 1 October to 30 April,
which is usually the heating period of city districts. The results reveal that at a depth level of 20 m (not shown), the water was cold in every winter. Even on the mildest winter 2007/2008, water with T < 2 °C was present for several days. Going further to the deeper layers of 50, 60 and 70 m, the limit T < 2 °C is normally not found (except for very cold winters 1993/1994, 1994/1995, 1999/2000 and 2010/2011), and even cases with T < 3 °C were rather rare.

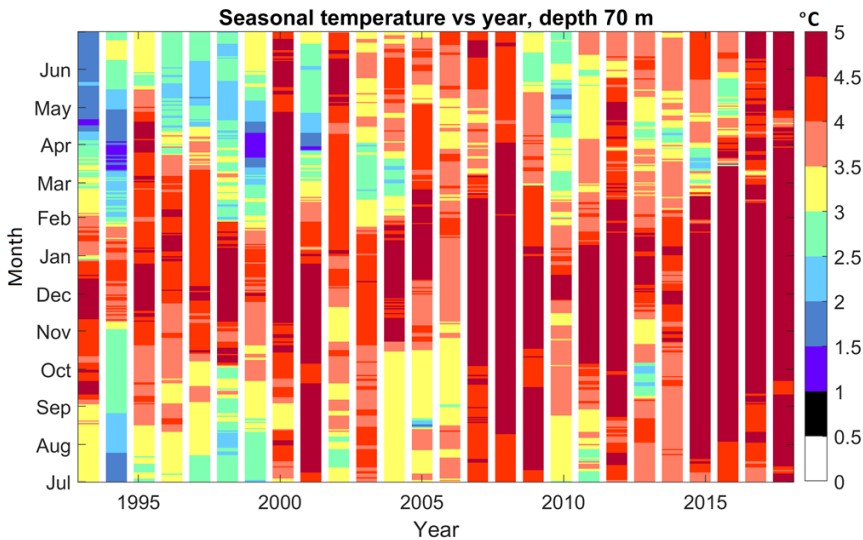

**Figure 6: Temperature diagrams in Tallinn Bay at 70 m depth as a function of year (horizontal axis) and calendar day (vertical axis), calculated from Copernicus Marine reanalysis data (product ref. no. 1).**





Profiles of the mean number of non-cold days (when heat extraction is considered possible) reveal that favorable periods corresponding to different temperature limits are minimal in the surface layer down to 20 m (Fig. 7). Further, with increasing

depths, extractable seawater heat becomes more frequent. Significant increase of duration of the seawater-based heating occurs from 30 to 50 m depth: from 107 to 166 days for 3 °C and 149 to 200 days for 2 °C. Considering the recent period 2009–2019, selected to cover the positive temperature trend, the mean duration non-cold days was generally by 5–10 days longer than for the whole period 1993–2019, whereas the largest increase up to 20 days was found at the deepest layers 60 and 70 m.

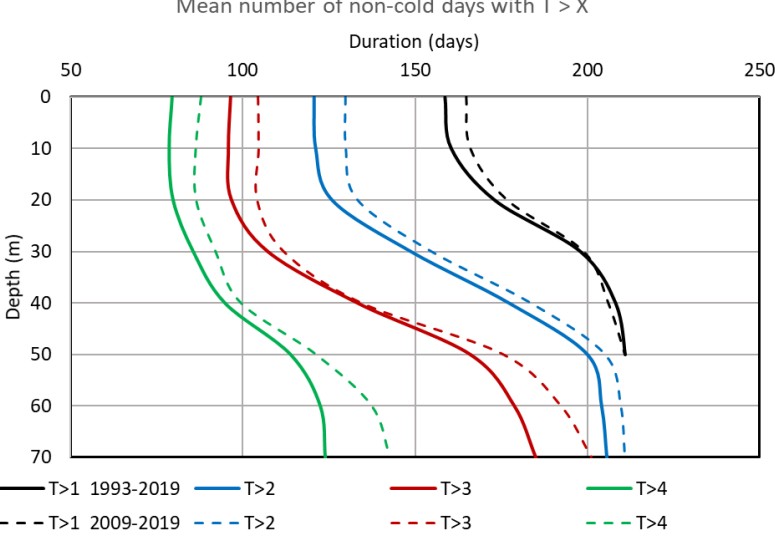

**Figure 7: Profiles of mean number of non-cold days for the reanalysis point 6 (Fig. 2). X = 1, 2, 3 and 4 °C as shown in the legend. Copernicus Marine reanalysis data from 1993–2019 for the full period (solid line) and for the recent 11 years 2009–2019 (dashed line). Data taken from 1 October to 30 April (211 days) of each year. Product ref. no. 1.**

Profiles of mean start dates of cold periods, when additional heating has to be switched on, were calculated from 1 November

of each year (Fig. 8). Both in the offshore and the coastal waters the start dates are nearly the same. The temperature goes below a certain limit first at the surface layer, and later in the deeper layers. Seawater becomes less than 3 °C on the average on 1 January at 20 m depth and on 12 February at 50 m depth. At the 70 m depth, the average start of T < 3 °C was calculated 28 February, although only 14 winters out of 26 had such water present; in 12 winters there was always the condition T > 3 °C fulfilled. We note that calculations made with the mean seasonal cycle (Fig. 4) revealed always T > 3 °C. During the recent

warmer period 2009–2019, the start has been delayed on the average by 5–10 days. However, this estimation is rather uncertain, since there have been several winters where the criteria T < 1 °C, T < 2 °C and T < 3 °C have not been met due to dominating warmer waters. Among the winters in 1993–2019, temperature at 50 m depth was more than 5 days above 3 °C during 9 winters (35%) and above 2 °C during 18 winters (70%). The coldest winters as shown in Fig. 6 with low deep-water temperature were 1993/1994, 1994/1995, 1999/2000 and 2010/2011.






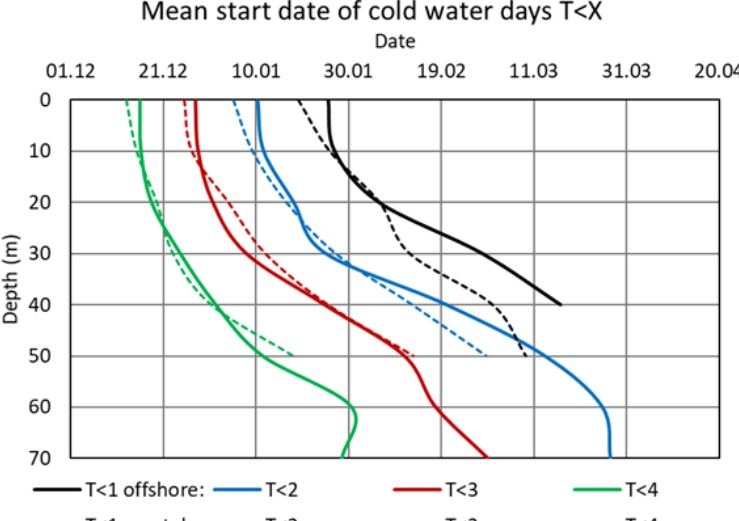

**Figure 8: Profiles of mean start date of cold seawater period. X = 1, 2, 3 and 4 °C as shown in the legend. Copernicus Marine reanalysis data (product ref. no. 1) from 1993–2019 from 1 October to 30 April in the offshore point 6 (solid line) and internal bay point 3 (dashed line) Locations of points are given in Fig. 2.**


The results in Fig. 8 show that statistical properties of cold/warm water occurrence are in the first approach horizontally uniform, i.e. average data from one location can be extended over the entire bay area. Since seawater intake for the heat pump systems is usually located on the bottom, then in the small bay its baseline temperature regime can be characterized from the temperature time series of nearby deeper location.


### 3.3 Frequency of extractable seawater heat occurrence over the Baltic Sea bottom

Seawater heat pumps can use the water with temperature above a certain limit; adopt here $T > 3.5$ °C for non-cold water. Since intake tubes are located on the bottom, "good" locations for seawater heat pump installations are those where non-cold bottom waters $T_{bot} > 3.5$ °C occur most frequently, leaving a minimum number of days when an additional (usually, less effective)
energy source is needed.

Copernicus Marine reanalysis data were used to calculate Baltic-wide non-cold water occurrence frequency on the bottom, using the data from the heating period (from October to April) during 1993–2021 (Fig. 9). At each grid cell of the reanalysis data, the near-bottom temperature value was each day compared to the non-cold criteria; and the counted number of non-cold
days was divided by the full number of days during the period, resulting $F = 100\%$ for the regions where always $T_{bot} > 3.5$ °C.



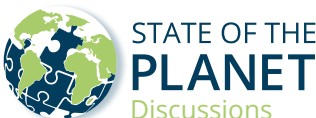

We may adopt that seawater heat is extractable in the regions with $F > 85\%$ as shown by color scale in Fig. 9. Note that extractable seawater heat is found in the deeper basins of the Baltic Proper, where permanent halocline exists (Fig. 1).

From the socio-economic viewpoint, favorable locations for seawater heat extraction combine in a short distance the regions
of extractable seawater heat with the significant urban and/or industrial areas. Among larger cities, Tallinn has one of the most favorable locations for using the seawater heat since the boundary for $F > 85$ % lies just a few km from the coast. In terms of extractable marine heat distance only, the best locations in the Baltic Proper are the western coast of Gotland Island of Sweden and the tip of Hel Peninsula near TriCity of Poland (Gdansk, Sopot, Gdynia) but these regions have only moderate coastal population. Good locations for seawater heat extractions are found near Copenhagen and other Danish cities, and TriCity of
Poland.

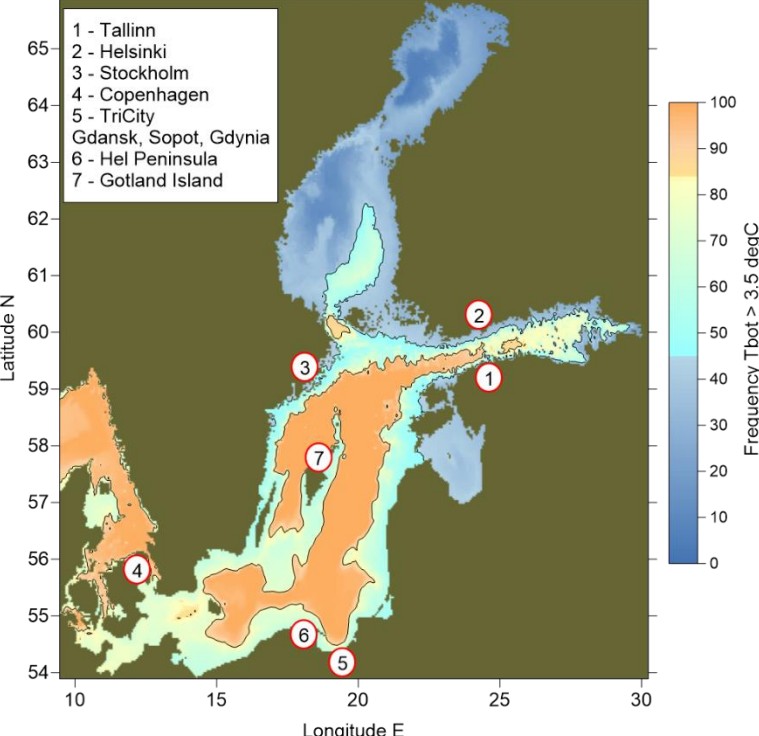

**Figure 9: Frequency of occurrence of non-cold waters $T_{bot} > 3.5$ °C, calculated from the Copernicus Marine reanalysis data 1993–2021 (product ref. no. 1).**

## 4. Discussion and outlook

The design of seawater heat pumps is a complicated interdisciplinary optimization task (Schibuola et al., 2022). There should be heating or cooling energy users in a reasonable distance from the coast, since a fraction of gained energy is lost to the pumping. In the ice-prone (boreal) seas, there should be locations near the large-scale users where seawater temperature by





depth does not frequently fall below +3 °C. In most cases, it is useful to combine seawater heat energy with other renewable sources, such as groundwater, wastewater, lakes, rivers and air (Pieper et al., 2022; Volkova et al., 2022b).


Favorable oceanographic conditions are of crucial importance for the planning and operation of seawater heat pumps. Important factors include local bathymetry, water temperature, peak heating load, and screening requirements i.e. to prevent the entrainment of biological organisms and other particulate matter into the system (Mitchell and Spitler, 2013). The present pilot study used long-term and regular, but coarse reanalysis data to determine the near-bottom seawater temperature

characteristics in the context of potential use of seawater thermal energy for district heating. Heat pumps are operated continuously in automated regime (although there is a response time depending on the system characteristics). Seawater intake systems have limited horizontal extent (much smaller than 3.7 km, the grid step of present reanalysis), therefore unresolved fine-scale coastal and topographic features (especially slope effects, e.g. Delpeche-Ellmann et al., 2018) may become important in generation of meso- and submesoscale dynamics and related temperature variations.


In heat engineering, statistics of various meteorological factors (air temperature, wind, solar radiation etc) is synthesized in the concept of "energy reference year" containing hourly forcing (Kalamees et al., 2012). It allows design of different heat system components based on patterns of heat production and consumption. The reference year is composed from the monthly segments from different years of meteorological data that "clueing" should in the best way present the statistical distributions over the

longer climatic period. For the design of seawater components of the overall district heating system it could be useful to compose also the oceanographic reference years. This is not a trivial task, since many aspects have to be kept in mind. Oceanographic reference year should match seawater conditions; therefore, the best segment years do not need to correspond to the years for best meteorology statistical match. Since VHR data at the pumping sites are usually not available, then modeling of such data requires that the local forcing data and that of the adjacent sea area agree, that contradicts to the piece-

wise combination of meteorological reference data. Harmonizing the meteorological and oceanographic reference data for heat and energy engineering is a topic of joint ongoing studies of natural and engineering scientists.

We have performed basic coarse-scale seawater temperature statistics from Copernicus Marine reanalysis and available observations. If evaluations by the energy companies indicate that using seawater heat seems technically and economically

feasible, the further studies of the oceanographic component of large-scale initiative should include the following steps.

1) Refine the statistics (including currents etc.), using results from the sub-regional forecast/reanalysis model with higher resolution. There is an operational forecast model with 1-km resolution (Lagemaa et al., 2011) running from 2009. Corresponding reanalysis is in the implementation phase.

2) Together with engineers, identify potential locations of seawater intake/discharge or heat exchanger etc. Conduct

dedicated observations and VHR modeling at selected locations.





3) Synthesize data from (1)–(2) and Copernicus Marine reanalysis into the custom-tailored data products, necessary for engineering designs. These data products could contain refined "reference year" time series in present and future climate, and event-based statistics adjusted to the operation decisions of the heat pumps.

4) Conduct also other studies (seabed habitats, sediments, bathymetric and geological data) both in the intake and
pipeline locations, needed for the environmental impact assessment.

## 5. Conclusions

Due to the growing interest of seawater heat extraction, variability of temperature in the Tallinn Bay was studied using the Baltic Sea reanalysis data 1993–2019 from the Copernicus Marine Service. The reanalysis data match the coastal temperature observations with bias -0.12 °C and RMSD (root-mean-square difference) of 1.3 °C.


During summer, upper mixed layer located above sharp thermocline, has typically 20 m thickness; its yearly temperature maximum varies between 16 and 23 °C. In autumn, the upper layer is eroded down to 40 m where inherent salinity stratification blocks further erosion, or down to the bottom, whichever is shallower. During winter, the top of the upper layer can reach the freezing point, which is approximately -0.3 °C. At a depth of 40 m, the monthly mean temperature ranges from 2.4 to 7.4 °C,

while at 70 m depth, it ranges from 3.4 to 5.2 °C. The highest temperatures at greater depths are typically observed in November and December, while the lowest values are commonly found on average in March and April.

Oceanographic conditions for seawater heat extraction are the least favorable in surface waters down to 20 m. Further, with increasing depths, extractable seawater heat becomes more frequent. Significant increase of duration of the seawater-based

heating occurs from 30 to 50 m depth: from 107 to 166 days for 3 °C and 149 to 200 days for 2 °C. Seawater becomes less than 3 °C on the average on 1 January at 20 m depth and on 12 February at 50 m depth.

Among larger cities, Tallinn has one of the most favorable locations for using the seawater heat. Good locations for seawater heat extractions are found near Copenhagen and other Danish cities, and TriCity of Poland.


## Data availability

Data are available from open sources as given in Table 1.



**Author contribution**

JE acted as a coordinating author. All the authors analyzed the data, and wrote and edited the paper.

**Special issue statement**

The paper belongs to the 8th edition of the Copernicus Marine Service Ocean State Report (OSR 8).

**Competing interests**

The contact author has declared that none of the authors has any competing interests.

**Acknowledgements**

The study was driven by the practical implementation interests of district heating and cooling, expressed by a company OÜ Utilitas, Geological Survey of Estonia and managers of a new climate neutral Hundipea district in Tallinn. Cooperation within Baltic Monitoring and Forecasting Centre of Copernicus Marine Service is highly appreciated.

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





**Table 1. Copernicus Marine Service and non-Copernicus products used in this study, including information on data documentation.**

| Product ref. no. | Product ID & type | Data access | Documentation |
|---|---|---|---|
| 1 | BALTICSEA_REANALYSIS_PHY_003_011; Numerical models | EU Copernicus Marine Service Product (2021); | Quality Information Document (QUID): Liu et al. (2019); Product User Manual (PUM): Axell et al. (2021) |
| 2 | INSITU_BAL_PHYBGCWAV_DISCRETE_MYNRT_013_032; In-situ observations | EU Copernicus Marine Service Product (2022) | Quality Information Document (QUID): Wehde et al. (2022); Product User Manual (PUM): In Situ TAC partners (2022) |
| 3 | EMODNET; bathymetry | EMODnet (2021) | EMODnet Bathymetry Consortium, 2018; Jakobsson et al. (2019) |
| 4 | OpenStreetMap; coastline | HELCOM MADS (2023) | OpenStreetMap Contributors (2023) |