# Peer review of "Oceanographic preconditions for planning seawater heat pumps in the Baltic Sea – an example from the Tallinn Bay, Gulf of Finland"

_State of the Planet, 2023_

## Author Comment (AC1)

Author's Response

Title: Oceanographic preconditions for planning seawater heat pumps in the Baltic Sea – an example from the Tallinn Bay, Gulf of Finland

Author(s): Jüri Elken et al.

MS No.: sp-2023-21

Report: 8th edition of the Copernicus Ocean State Report (OSR8)

**Referee#1**

The manuscript „Oceanographic predictions of…" is an interesting study for some oceanographic predictions of water temperature for the Tallin Bay with the view of use the sea water as a renewable source of energy.

The statistic use of data is valid and show some potential for that kind of use. However I have some comments about it. These comments are as follows:

Comment

*The title is "Oceanographic predictions…". While there are hardly any predictions within the manuscript. There is a proper statistical explanation of the reanalysis data but without predictions for future years of months. Therefore I would suggest to change slightly the title to avoid misunderstanding. Or go further with the topic and include some predictions. It is interesting that there is an increase of the Baltic Sea's temperatures since for example 2015/2016. Should it be categorized as a trend and can it be expected for further increase?*

Response

The user partners, mentioned in the acknowledgements, showed an interest to obtain heat-engineering-oriented seawater temperature statistics both for the historical period and for the coming decades. In this study we had to confine to the historical period only, based on Copernicus Marine Service reanalysis data from 1993-2019. Predictions were not included in the MS. The title has the formulation "Oceanographic preconditions for planning seawater heat pumps in the Baltic Sea – an example from the Tallinn Bay, Gulf of Finland".

Comment

*The manuscript aims to help with the planning of seawater heat pumps. However, there is very little about that kind of heat pumps in the manuscript. I suggest to add some details about seawater heat pumps technology, show some location where that kind of heat pumps works (not only are planned or analyzed) and write about some general conditions of seawater or climate.*

Response

Regarding existing heat pumps, in line 33 there is a reference to the heat pump in Ropsten, Stockholm. ACTION: The description in line 33 will be extended to show that the system is in operation since 1986. In addition to the Ropsten facility, reference to the new Danish large heat pump facility in Esbjerg https://www.man-es.com/discover/esbjerg-heat-pump will be also mentioned and referenced. Brief description of seawater heat pump principles will be added in the introduction starting from line 26.

Comment

*Page 2, L.31: The heat pump itself can work with fluids with temperature below 0ºC (depends on technology). The problem with water in the freezing point is rather for piping and transfer system of the water.*

Response

Based on our cooperation with heat engineers, keeping the water temperature above freezing point is important for the used or usable technologies. This MS is not planned to go into details of heat engineering; therefore, we would prefer the general wording as in line 31.

Comment

*Please, add units for salinity in Figure 1.*

Response

ACTION: Units will be added.

Comment

*Figure 1 b). How much stratification in the Gotland Deep refers to conditions of the Tallin Bay considered within this manuscript?*

Response

Figure 1b presents general background and definition of terms for the Central Baltic stratification. Specific features of the Gulf of Finland, in the context of the Gotland Basin, are described in lines from 74 to 114. ACTION: details for the Tallinn Bay stratification will be added, when appropriate.

Comment

*Data (2.2.1) – in my opinion the short explanation of how the data was achieved is needed. Was it a general data base or a combination of independent data sets which need to be for example validated?*

Response

ACTION: In line 136 the sentence "This reanalysis covers the whole Baltic Sea with adjacent North Sea areas." will be extended to list the variables available in the product data set. In line 145, after the sentence "The largest RMSD values were found in the Kattegat and the Gulf of Finland" we shall add a forward reference to the sub-chapter 2.3 where comparison of reanalysis data with the SST observations in Tallinn Port is presented.

Comment

*P6 L.146: Please add values od RMSD for Gulf of Finland*

Response

ACTION: Numeric RMSD values will be inserted into the sentence "The largest RMSD values were found in the Kattegat and the Gulf of Finland."

Comment

*Figure 2. If the water transport originated in North Sea and transported via Danish Straits, then why grid cells were taken from the North only? And without western grid cells. Can it influence the outcomes? Figure 2 also needs some general map for location of this area.*

Response

The Tallinn Bay box is already shown in Fig. 1a but it may be not noticed. ACTION: a larger scale map is added in Fig. 2.

Comment

*Figure 4. What is point 6? Is this the location grid cell from Figure 2 or number of reanalysis data? It needs to be highlighted? In the Figure 4 there is temperature and salinity given as an average for those 6 grid cells from Figure 2 or from a certain (representative) point?*

Response

When describing vertical variations, we take one offshore point (lines 194-195 say "In the offshore area (point 6, Fig. 2)…", also Fig. 4 legend says "reanalysis point 6 (Fig. 2)…". When interested in horizontal variations at selected depths (10 m and 50 m), we analyze all the data from 6 reanalysis points, Fig. 5. ACTION: The sentence starting in line 215 "Although the mean seasonal temperature curves are rather similar in the selected six reanalysis points (locations in Fig. 2), …" is rewritten as "Although in the selected six reanalysis points (locations in Fig. 2), the mean seasonal temperature curves are rather similar (comparison not shown),…".

Comment

*The analysis are provided for several depths (for example 10, 50, 70m). For heat pump technology not only this is important. Very important would be also distance from the heat pump location and the length of the piping system which will provide the water from certain depth with certain temperature. It would develop temperature difference between water from greater depths and shallower layers (or air temperature) and led to heat losses.*

Response

We have mentioned the distance to the coast in the abstract (lines 10, 19), and discussion and conclusions (lines 304, 307, 316). When seawater temperature is acceptable for intake, then the main question is in what shortest distance appropriate seawater is found. We have also noted (lines 316-317) that "fraction of gained energy is lost to the pumping".

Comment

*In my opinion, huge advantage of this manuscript would be to give some general statistics about air temperature in the area (for example mean monthly air temperature)*

Response

ACTION: We shall add line graph of air temperature and sea surface temperature to Fig. 4.

Comment

*Figure 7. In my opinion the T>1 curve need short explanation why it ends at 50 m.*

Response

This is an error due to recalculations from the earlier alternative presentation. Below 50 m the water is always warmer than 2 degC. ACTION: Fig 7 will be corrected, maximum number of non-cold days (211) will be also shown.

Comment

*When considering exploitation the water of temperature about 1ºC for heat pumps some inconvenience of that solution must be taken into account. For example, there would be extraction of water of near freezing temperature in big amounts back into the seawater. Would it change the temperature stratigraphy and provide anthropogenic water mixing? Therefore the location is quite crucial for heat pump installations.*

Response

In the discussion (lines 345-355) we have outlined the need to conduct studies on currents, temperature and salinity, seabed habitats, sediments, bathymetric and geological data both in the intake and pipeline locations, needed for the environmental impact assessment. We have considered temperature of intake water at least 3 degC. Outfall of cooled waters may stabilize the water column when temperature is below 2.5 deg (approximately temperature of maximum density). Another hydrodynamic problem arises when saltier water is pumped from deep offshore region and released in coastal area where ambient water is less saline. This may create anthropogenic plume of deeper water and associated coastal jet. When deeper water contains more nutrients than coastal water, then the pump system could act as nutrient source for the coastal waters. The impact question is rather multifaceted and complex. Therefore we had to limited the scope of discussion.

Comment

*Figure 9. should be accompanied with the separate map (or information included in the Figure 9) about the depth where T>3,5ºC is valid.*

Response

ACTION: Depth contours will be added to Fig. 9 by dashed lines.

---

## Author Comment (AC2)

Author's Response

Title: Oceanographic preconditions for planning seawater heat pumps in the Baltic Sea – an example from the Tallinn Bay, Gulf of Finland

Author(s): Jüri Elken et al.

MS No.: sp-2023-21

Report: 8th edition of the Copernicus Ocean State Report (OSR8)

**Referee#2**

Comment
*The article complies with the good practice of scientific articles. However, there is little scientific innovation. Rather, we get updated and detailed information about changes in seawater temperature.*

Response
For ourselves, a new aspect was to go beyond the traditional statistics of monthly means, variance etc., meeting the interests of end users.

Comment
*The title of the article "Oceanographic preconditions...", just this word "preconditions" corresponds to the content of the article. A heat engineer must answer whether the heat pump is suitable to be placed in Tallinn Bay. The article provides information for this.*

Response
We agree

Comment
*I recommend publishing the article as presented*

---

## Author Response (AR1)

**Point-by-Point Author's Response & Manuscript Revision**

Title: Oceanographic preconditions for planning seawater heat pumps in the Baltic Sea – an example from the Tallinn Bay, Gulf of Finland

Author(s): Jüri Elken et al.

MS No.: sp-2023-21

Report: 8th edition of the Copernicus Ocean State Report (OSR8)

Changes have been done using "track changes". The changes are commented in the revised text using abbreviations from C#2 to C#14.

**Referee#1**

The manuscript „Oceanographic predictions of…" is an interesting study for some oceanographic predictions of water temperature for the Tallin Bay with the view of use the sea water as a renewable source of energy.

The statistic use of data is valid and show some potential for that kind of use. However I have some comments about it. These comments are as follows:

**Comment#1**

*The title is "Oceanographic predictions…". While there are hardly any predictions within the manuscript. There is a proper statistical explanation of the reanalysis data but without predictions for future years of months. Therefore I would suggest to change slightly the title to avoid misunderstanding. Or go further with the topic and include some predictions. It is interesting that there is an increase of the Baltic Sea's temperatures since for example 2015/2016. Should it be categorized as a trend and can it be expected for further increase?*

**Response#1**

The user partners, mentioned in the acknowledgements, showed an interest to obtain heat-engineering-oriented seawater temperature statistics both for the historical period and for the coming decades. In this study we had to confine to the historical period only, based on Copernicus Marine Service reanalysis data from 1993-2019. Predictions were not included in the MS. The title has the formulation "Oceanographic preconditions for planning seawater heat pumps in the Baltic Sea – an example from the Tallinn Bay, Gulf of Finland".

The subsurface temperature trends have been derived from Copernicus Marine Service reanalysis data from 1993-2022 for the Baltic Sea (DOI:10.48670/moi-00208). Horizontal averaging has been done over the Baltic Sea domain (13 °E - 31 °E and 53 °N - 66 °N). The results show warming trends of about 0.05 °C/year at all depths.

**Comment#2**

*The manuscript aims to help with the planning of seawater heat pumps. However, there is very little about that kind of heat pumps in the manuscript. I suggest to add some details about seawater heat pumps technology, show some location where that kind of heat pumps works (not only are planned or analyzed) and write about some general conditions of seawater or climate.*

**Response#2**

Regarding existing heat pumps, in line 33 there is a reference to the heat pump in Ropsten, Stockholm.

ACTION: The description in line 33 has been extended to show that the system is in operation since 1986. In addition to the Ropsten facility, reference to the new Danish large heat pump facility in Esbjerg https://www.man-es.com/discover/esbjerg-heat-pump will has been mentioned and referenced.

Brief description of seawater heat pump principles has been added in the introduction starting from line 26.

One energy source for the heat pumps is seawater (Bach et al., 2016) that has stable temperature during the winter, compared to the air temperature. Seawater heat pumps take seawater from the locations of appropriate temperature into the shore unit that transforms its low temperature to higher temperature suitable for the district heating.

To describe seawater heat pumps technology in detail is out of the scope of this manuscript. An example of the engineering approach of planning heat pumps for district cooling in Tallinn is described in Volkova et al. (2022a) (this reference is in the MS).

**Comment#3**

*Page 2, L.31: The heat pump itself can work with fluids with temperature below 0ºC (depends on technology). The problem with water in the freezing point is rather for piping and transfer system of the water.*

**Response#3**

Based on our cooperation with heat engineers, keeping the water temperature above freezing point is important for the used or usable technologies. This MS is not planned to go into details of heat engineering; therefore, we would prefer the general wording as in line 31.

**Comment#4**

*Please, add units for salinity in Figure 1.*

**Response#4**

Salinity unit has been added.

**Comment#5**

*Figure 1 b). How much stratification in the Gotland Deep refers to conditions of the Tallin Bay considered within this manuscript?*

**Response#5**

Figure 1b presents general background and definition of terms for the Central Baltic stratification. Specific features of the Gulf of Finland, in the context of the Gotland Basin, are described in lines from 74 to 114.

ACTION: New paragraph about the stratification processes has been added.

Tallinn Bay is deep enough to have all the above-described water column layers present. Its short-term hydrographic variability is characteristic to the southern coast of the Gulf of Finland: upwelling occurs during the persistent westerly and southwesterly winds; pulses of more saline and less saline water observed in the southern Gulf of Finland appear also in the bay.

**Comment#6**

*Data (2.2.1) – in my opinion the short explanation of how the data was achieved is needed. Was it a general data base or a combination of independent data sets which need to be for example validated?*

**Response#6**

ACTION: In line 136 the sentence "This reanalysis covers the whole Baltic Sea with adjacent North Sea areas." Has been extended to list the variables available in the product data set. In line 145, after the sentence "The largest RMSD values were found in the Kattegat and the Gulf of Finland" a forward reference has been added to the sub-chapter 2.3 where comparison of reanalysis data with the SST observations in Tallinn Port is presented.

In the Tallinn Bay, the reanalysis results will be compared to the independent sea surface temperature observations in the sub-section 2.3.

**Comment#7**

*P6 L.146: Please add values od RMSD for Gulf of Finland*

**Response#7**

ACTION: Numeric RMSD values have been inserted.

**Comment#8**

*Figure 2. If the water transport originated in North Sea and transported via Danish Straits, then why grid cells were taken from the North only? And without western grid cells. Can it influence the outcomes? Figure 2 also needs some general map for location of this area.*

**Response#8**

ACTION: a larger scale map of the Gulf of Finland has been added in Fig. 2.

**Comment#9**

*Figure 4. What is point 6? Is this the location grid cell from Figure 2 or number of reanalysis data? It needs to be highlighted? In the Figure 4 there is temperature and salinity given as an average for those 6 grid cells from Figure 2 or from a certain (representative) point?*

**Response#9**

When describing vertical variations, we take one offshore point (lines 194-195 say "In the offshore area (point 6, Fig. 2)...", also Fig. 4 legend says "reanalysis point 6 (Fig. 2)...". When interested in horizontal variations at selected depths (10 m and 50 m), we analyze all the data from 6 reanalysis points, Fig. 5.

ACTION: The sentence starting in line 215 "Although the mean seasonal temperature curves are rather similar in the selected six reanalysis points (locations in Fig. 2), ..." has been rewritten:

Although in the selected six reanalysis points (their locations are in Fig, 2) the mean seasonal temperature curves are rather similar (Fig. 5)), the spread of the actual data is rather high.

**Comment#10**

*The analysis are provided for several depths (for example 10, 50, 70m). For heat pump technology not only this is important. Very important would be also distance from the heat pump location and the length of the piping system which will provide the water from certain depth with certain temperature.*

*It would develop temperature difference between water from greater depths and shallower layers (or air temperature) and led to heat losses.*

Response#10

We have mentioned the distance to the coast in the abstract (lines 10, 19), and discussion and conclusions (lines 304, 307, 316). When seawater temperature is acceptable for intake, then the main question is in what shortest distance appropriate seawater is found. We have also noted (lines 316-317) that "fraction of gained energy is lost to the pumping".

**Comment#11**

*In my opinion, huge advantage of this manuscript would be to give some general statistics about air temperature in the area (for example mean monthly air temperature)*

**Response#11**

ACTION: We have added a line graph of air temperature and sea surface temperature in Fig. 4.

**Comment#12**

*Figure 7. In my opinion the T>1 curve need short explanation why it ends at 50 m.*

**Response#12**

This is an error due to recalculations from the earlier alternative presentation. Below 50 m the water is always warmer than 2 °C.

ACTION: Fig 7 has been corrected.

**Comment#13**

*When considering exploitation the water of temperature about 1ºC for heat pumps some inconvenience of that solution must be taken into account. For example, there would be extraction of water of near freezing temperature in big amounts back into the seawater. Would it change the temperature stratigraphy and provide anthropogenic water mixing? Therefore the location is quite crucial for heat pump installations.*

**Response#13**

In the discussion (lines 345-355) we have outlined the need to conduct studies on currents, temperature and salinity, seabed habitats, sediments, bathymetric and geological data both in the intake and pipeline locations, needed for the environmental impact assessment. We have considered temperature of intake water at least 3 °C. Outfall of cooled waters may stabilize the water column when temperature is below 2.5 °C (approximately temperature of maximum density). Another hydrodynamic problem arises when saltier water is pumped from deep offshore region and released in coastal area where ambient water is less saline. This may create anthropogenic plume of deeper water and associated coastal jet. When deeper water contains more nutrients than coastal water, then the pump system could act as nutrient source for the coastal waters. The impact question is rather multifaceted and complex. Therefore, we had to limit the scope of discussion.

**Comment#14**

*Figure 9. should be accompanied with the separate map (or information included in the Figure 9) about the depth where T>3,5ºC is valid.*

**Response#14**

ACTION: Depth contour 50 m has been added to Fig. 9.